# Effectiveness of brief mindfulness intervention for college students' problematic smartphone use: The mediating role of self-control

**Fengbo Liu** [1]*, **Zhongqiu Zhang**[2], **Shuqiang Liu**[3], **Zhantao Feng**[4]

1 School of Physical Education, Zhengzhou University of Light Industry, Zhengzhou, Henan Province, China, 2 China Institute of Sport Science, General Administration of Sport of China, Beijing, China, 3 School of Psychology, Beijing Sport University, Beijing, China, 4 College of Sports and Health, Shandong Sport University, Rizhao, Shandong Province, China

* lfbzzuli@126.com

## Abstract

### Background

Mainland China has the most smartphone users worldwide, especially among college students, while mindfulness intervention can significantly alleviate the level of problematic smartphone use. We examined the effects of a brief mindfulness intervention on problematic smartphone use and investigated if this effect is mediated by self-control.

### Methods

Participants were recruited randomly from a university in Beijing of China. Forty-four college students were assigned to a mindfulness group or a control group. The mindfulness group took part in a brief (30 min) single-session mindfulness intervention. The control group was instructed to listen to a neutral news audio recording for the same duration (30 min). The Freiburg Mindfulness Inventory, Mobile Phone Addiction Tendency Scale, and Self-control Scale were used to measure state mindfulness, problematic smartphone use, and self-control of college students at pre-intervention and post-intervention, respectively.

### Results

Two-way repeated-measures ANOVA revealed that the mindfulness group had significant improvements in state mindfulness ($p = .049$) and self-control ($p = .012$), and had significant alleviation in problematic smartphone use ($p < .001$) at post-intervention. In the regression model, self-control had a mediating effect between mindfulness intervention and problematic smartphone use (95% *CI* [0.490, 7.216]).

### Conclusions

A brief single-session mindfulness intervention can alleviate the level of problematic smartphone use and increase the level of state mindfulness and self-control compared to the

**Data Availability Statement:** The [DATA.sav] data used to support the findings of this study are included within the supplementary information file.

**Funding:** This study was supported by a Doctor Research Fund (2022BSJJSK12) from Zhengzhou University of Light Industry. The funder provided support for the preparation and publication of this paper. It was also supported by Zhengzhou University of Light Industry for providing the psychological laboratories, and Beijing Sport University for providing the participants.

**Competing interests:** The authors have declared that no competing interests exist.

control group. Self-control can completely mediate the efficacy of the mindfulness intervention in reducing problematic smartphone use.

## 1 Introduction

Smartphones are multipurpose devices in modern life which provide substantial convenience, but problematic smartphone use is becoming a serious problem and is increasingly prevalent worldwide [1]. According to data from recent surveys, 39%~44% of India adolescents tend to have problematic cell phone usage [2]. A large UK cross-sectional study found that 39% of young adults reported smartphone addiction [3], whereas a study in America reported that 25% of people had a smartphone addiction problem [4]. Because of its extremely large population base, Mainland China has the most smartphone users worldwide, especially college students [5].

Although the smartphone brings convenience to people's daily lives, it is also associated in certain cases with patterns of addictive usage involving negative outcomes. A large number of studies have introduced the term "smartphone addiction" based on similarities in symptoms displayed by excessive smartphone users and substance abusers [6]. For example, problematic use of smartphones can cause physical and mental health symptoms (e.g., decline in academic performance [7], interpersonal communication barriers [8], even threaten their life safety [9]) for college students. Therefore, how to effectively prevent and intervene with problematic smartphone use among college students has become an important research topic now.

Both theory and practice show that psychotherapy is effective in alleviating the level of problematic smartphone use among college students [10]. Psychotherapy is defined as the treatment of mental health symptoms or disorders or problems of living, and/or facilitation of personal growth by psychological means [11]. Psychotherapy, with or without pharmacologic therapy, is effective for the treatment of mental health symptoms and disorders [12]. Individual psychotherapy, especially cognitive-behavioral therapy (CBT), is efficacious for treating problematic smartphone use in the general population [13].

The current emerging CBT is represented by "mindfulness training". Mindfulness means that the individual keeps his attention on purpose and does not make judgments (nonjudgement) in the present moment, and to be aware of current events [14], that is, stay alert and focus on the present event at all times, and be ready to accept any possible situation without judgment [15]. Similarly, mindfulness training is based on acceptance, which is an intervention method for individuals to focus their attention on the present experience [16]. Studies have shown that mindfulness intervention can significantly alleviate the level of problematic smartphone use among college students. For example, Elhai recruited 261 college students for a repeated-measures web survey, and found that mindfulness was inversely associated with levels of problematic smartphone use [17]; Yang recruited 1258 students in China for a questionnaire survey, and found that the relationships between smartphone addiction and mental health symptoms were moderated by mindfulness, in which they were stronger for students with lower levels of mindfulness [18]; Lan recruited 70 university students with smartphone addiction, and they were divided into a control group ($n = 29$) and an intervention group ($n = 41$), which demonstrated that the group mindfulness-based cognitive-behavioral intervention could significantly alleviate smartphone addiction [19].

Although previous studies have preliminarily confirmed the effect of mindfulness intervention on problematic smartphone use [19], its mediating mechanism has not been fully

revealed. And most existing studies focused on the mediating effect of trait mindfulness [20], rather than the mediating mechanism of mindfulness intervention. Therefore, it is necessary to further supplement the mediating mechanism of mindfulness intervention on mobile phone addiction. After combing through previous studies, it is found that when discussing college students' mindfulness intervention and problematic smartphone use, it is often associated with self-control ability. Self-control refers to the ability to surpass or change a person's internal reactions, including the ability to consciously interrupt the flow of thought, change emotions, and suppress unwelcome impulsive thoughts and behaviors [21]. Previous studies indicated that mindfulness intervention can improve an individual's self-control [22], and a brief mindfulness intervention is also effective [23], because the concept of mindfulness is not to react, not to judge, and to pay attention to the present. Mindfulness intervention requires individuals to maintain their attention on the awareness of current events and need to deal with distractions caused by irrelevant thoughts, which can be seen as self-control training, where self-control ability is repeatedly and continuously practiced and significantly improved.

Besides, previous studies indicated that self-control has a positive predictive effect on some positive behaviors [24], or a negative predictive effect on some negative behaviors (such as problematic smartphone use or internet addiction, etc.) [25]. According to the dual-systems model of self-control proposed by Hofmann, self-control includes two opposing systems: the impulse system and the self-control system [26]. In the face of impulsive desires, the self-control system guides individuals to conduct thoughtful evaluation and inhibition standards, and then take rational actions. This system has the ability to inhibit psychological and behavioral impulsive responses [27]. Therefore, when college students' self-control ability improves, the alleviation of problematic smartphone use is more expected.

The majority of previous studies had focused on the effects of long-term mindfulness training on self-control and problematic smartphone use [28, 29], while a few intervention studies had explored the relationship between brief mindfulness training, self-control, and problematic smartphone use for college students; thus, the adoption of brief mindfulness training in this study is purposeful and fills a missing research gap. Clearly, there could be potential confounding factors arising from prolonged mindfulness interventions, such as improved interpersonal relationship [30], which makes it difficult for singling out mindfulness practice as the cause of improved problematic smartphone use. However, one-time brief intervention can avoid such problems. Thus, evidence of whether a brief mindfulness training affects the problematic smartphone use of college students reveals the effects more directly. Furthermore, some scholars highlighted the potential difficulty in getting college students to use mindfulness strategies effectively [31]. Long-term mindfulness training, for example, has been known to be easy for participants to drop out [32]. Currently, research evidence suggests the effectiveness of brief mindfulness training in eliciting positive outcomes, such as tolerance to negative affect [33]. The effects of such brief interventions led us to consider whether a brief mindfulness training would similarly improve college students' self-control and problematic smartphone use.

Previous studies have examined the changes in problematic smartphone use and self-control before and after mindfulness intervention [28, 29]. However, the mediating mechanism by which mindfulness intervention improves problematic smartphone use has rarely been explored. In addition, our efforts in testing the effects of brief mindfulness training over a short duration of 30 minutes hopefully contribute to the development of a simple strategy that can be readily applied to college students without expectation for prolonged sitting meditation. Therefore, the aim of the present study was to examine the effects of brief mindfulness training on college students' problematic smartphone use and its mediating mechanism. Based on the presented theoretical review we developed the following hypotheses: (1) brief mindfulness training can improve problematic smartphone use among college students; (2) brief

mindfulness training can improve self-control among college students; (3) self-control plays a mediating role between brief mindfulness training and problematic smartphone use; that is, brief mindfulness training can improve college students' problematic smartphone use through improving their self-control.

## 2 Methods

### 2.1 Participants

Ethics clearance was granted by the research ethics committee of the Beijing Sport University. All individuals were recruited online. Interested individuals were directed to an online screening questionnaire to determine eligibility. If individuals met inclusionary criteria, they were invited to participate in the study. Before the initiation of the study, written informed consent was provided to participants before inclusion and the confidentiality and anonymity of their participation were assured.

Lan reported a medium-to-large effect size of group mindfulness-based intervention for college students' problematic smartphone use [19]. An a priori power analysis determined that we would need a total sample size of 48 participants to detect this effect size (G*power; Effect size $f$ = .36, $\alpha$ = .05, 1 - $\beta$ = .80, Corr among rep measures = .50). In this study, the participants included 48 college students (15 boys and 33 girls; age average = 18.2, $SD$ = 1.5) from one college in Beijing of China. Participants were randomly separated into two groups: a mindfulness group and a control group. Due to time commitment, 22 of the 24 students in the mindfulness group (9 boys and 13 girls; age average = 18.6, $SD$ = 1.0) and 22 of the 24 students in the control group (4 boys and 18 girls; age average = 17.8, $SD$ = 1.8) completed the study.

### 2.2 Measures

Three instruments were used to assess college students' state mindfulness, level of smartphone addiction, and self-control ability: the Freiburg Mindfulness Inventory (FMI), Mobile Phone Addiction Tendency Scale (MPATS), and Self-control Scale (SCS).

State mindfulness was measured using the Chinese version of FMI [34] developed in a recent study [35] with supporting evidence regarding the factorial validity of the measure in a Chinese population. The measure has 14 items (e.g., "I am willing to accept the current experience."), and data from previous studies supported its factorial validity. Responses were made on 4-point Likert scale (1 = never; 4 = very often). This single-dimension scale produced satisfactory internal consistency during our assessments ($\alpha$ = .83).

Problematic smartphone use was measured using the Chinese version of MPATS [36]. Using a five-point Likert scale (1 = very rarely true; 5 = always true), participants rated 16 items from four dimensions, including withdrawal symptoms (e.g., "I feel uncomfortable if I don't use my phone for a long time."), highlighting behavior (e.g., "In class, I can't concentrate on listening because of phone calls or text messages."), social comfort (e.g., "I would rather choose to chat on my phone than face to face communication."), and mood change (e.g., "I am always afraid that my phone will turn off automatically."). Their responses were summed to create smartphone addiction scores. There was acceptable internal consistency reliability of the MPATS across our assessments ($\alpha$ = .91).

Self-control was measured using the Chinese version of SCS [21] developed in a recent study [37] with supporting evidence regarding the factorial validity of the measure in a Chinese population. Using a five-point Likert scale (1 = very rarely true; 5 = always true), participants rated 19 items from five dimensions, including impulse control (e.g., "People would describe me as impulsive."), healthy habits (e.g., "It is difficult for me to break bad habits."), resist temptation (e.g., "I lose my temper too easily."), focus on work (e.g., "I can work

effectively toward long-term goals."), and moderate entertainment (e.g., "I spend too much money.").Their responses were summed to create self-control scores. There was acceptable internal consistency reliability of the SCS across our assessments ($\alpha$ = .86).

### 2.3 Procedure

A 2×2 mixed factorial design was employed, with a group (mindfulness group vs. control group) as a between-subject factor, and time (pre-intervention vs. post-intervention) as a within-subject factor. All participants were tested individually. At baseline, each participant completed a brief demographic questionnaire to assess age, gender, and meditation experience. Afterward, each participant completed a self-report measure of mindfulness, problematic smartphone use, and self-control, and was then randomly assigned to one of the two groups.

Participants in the mindfulness group were seated in an empty classroom and were instructed to listen to a mindfulness training audio recording (30 minutes) [38], and complete the exercises outlined in the audio recording. The mindfulness training recording was recorded in advance by a mindfulness instructor who has more than 6 years of mindfulness intervention experience. The recording included 5-minute mindfulness breathing and 25-minute body scanning, which were traditional mindfulness exercises. Participants in the control group were instructed to listen to a neutral news audio recording for the same duration (30 minutes). Following the mindfulness or neutral recording, each participant completed a self-report measure of mindfulness, problematic smartphone use, and self-control.

### 2.4 Data analysis

Independent-sample t-tests were used to compare the demographic variables and the pre-intervention scores of mindfulness group and control group; Bivariate correlation was used to test the relationship between mindfulness, problematic smartphone use, and self-control; Two-way repeated-measures analyses of variance (ANOVA) were conducted to test the effect of time (within-subject independent variable: levels = pre-intervention and post-intervention) and group (between-subject independent variable: levels = mindfulness group and control group) on mindfulness, problematic smartphone use, and self-control, respectively; The macro program PROCESS [39] of SPSS compiled by Hayes was used to test the mediating effect of self-control between mindfulness intervention and problematic smartphone use. The number of Bootstrap samples was 5000. Under the 95% confidence interval, post-intervention score of problematic smartphone use were used as the dependent variable, the group was used as the independent variable, and the score change of self-control (post-intervention minus pre-intervention) was used as the mediating variable. Since the post-intervention score of problematic smartphone use were affected by pre-intervention score, therefore, gender, age, and pre-intervention score of problematic smartphone use were all controlled as covariates.

## 3 Results

### 3.1 Description results and bivariate correlation

The final sample included 44 college students. From Table 1, it can be found that there was no significant difference in both age ($t$ = 1.85, $df$ = 42, $p$ = .071) and gender ($t$ = -1.67, $df$ = 39.75, $p$ = .103) between the two groups.

From Table 2, it can be found that in pre-intervention, the mindfulness level of college students was negatively correlated with smartphone addiction ($r$ = —.30, $p$ < .05), and was positively correlated with self-control ($r$ = .51, $p$ < .01), that is, the higher level of mindfulness, the lower level of problematic smartphone use, and the higher level of self-control; In post-

**Table 1. Descriptive statistics and independent sample t-test comparison of the demographic variables.**

| variables | mindfulness group (n = 22) | control group (n = 22) | t | df | p |
|---|---|---|---|---|---|
| age | 18.64±0.95 | 17.82±1.84 | 1.85 | 42 | 0.071 |
| gender | 1.59±0.50 | 1.82±0.39 | -1.67 | 39.75 | 0.103 |

*Note.* For gender, 1 = male, 2 = female.

intervention, the mindfulness level was positively correlated with self-control ($r = .35$, $p < .01$), that is, the higher level of mindfulness, the higher level of self-control. The correlation between different variables provided a prerequisite for the subsequent mediation effect test.

### 3.2 Independent-sample t-tests results

It can be found that there was no significant difference in MPATS ($t = .52$, $df = 42$, $p = .609 > .05$) and SCS ($t = -1.58$, $df = 42$, $p = .121 > .05$) at pre-intervention between mindfulness group and control group, while there was significant difference in FMI ($t = -2.94$, $df = 42$, $p = .005 < .05$).

### 3.3 Two-way repeated measures ANOVA results

From Table 3, repeated measure ANOVA of mindfulness revealed a significant time by group interaction effect, $F(1, 42) = 6.68$, $p = .013$, $\eta_p^2 = .137$, and within-subject difference of mindfulness across time was not significant, $F(1, 42) = 0.08$, $p = .775$, $\eta_p^2 = .002$. Furthermore, between-group difference of mindfulness was not significant, $F(1, 42) = 2.89$, $p = .097$, $\eta_p^2 = .064$. Further simple effect analysis indicated that the mindfulness level of mindfulness group was significantly higher in post-intervention, as compared to that of pre-intervention ($p = .049$). See Fig 1.

Repeated measure ANOVA of problematic smartphone use revealed a significant time by group interaction effect, $F(1, 42) = 11.10$, $p = .002$, $\eta_p^2 = .209$, and within-subject difference of problematic smartphone use across time was significant, $F(1, 42) = 8.54$, $p = .006$, $\eta_p^2 = .169$. Furthermore, between-group difference of problematic smartphone use was not significant, $F(1, 42) = .28$, $p = .599$, $\eta_p^2 = .007$. Further simple effect analysis indicated that the problematic

**Table 2. Descriptive statistics and correlation test of the study variables.**

| variables | M | SD | FMI | MPATS | SCS |
|---|---|---|---|---|---|
| gender | 1.70 | 0.46 | | | |
| age | 18.23 | 1.51 | | | |
| pre-intervention | | | | | |
| FMI | 37.75 | 6.29 | - | | |
| MPATS | 41.80 | 11.88 | -0.30* | - | |
| SCS | 56.86 | 9.49 | 0.51** | -0.53** | - |
| post-intervention | | | | | |
| FMI | 38.00 | 6.75 | - | | |
| MPATS | 38.55 | 12.38 | -0.06 | - | |
| SCS | 57.91 | 8.93 | 0.35* | -0.46** | - |

*Note.* M = Mean; SD = Standard Deviation;

*$p < .05$

**$p < .01$; Average score of each variable is presented.

**Table 3. Summaries of two-way repeated-measures ANOVA comparison.**

| variables | within-subject | | | | | | between-subject | | |
|---|---|---|---|---|---|---|---|---|---|
| | time | | | time× group | | | $F$ | $p$ | $\eta_p^2$ |
| | $F$ | $p$ | $\eta_p^2$ | $F$ | $p$ | $\eta_p^2$ | | | |
| FMI | 0.08 | 0.775 | 0.002 | 6.68 | 0.013 | 0.137 | 2.89 | 0.097 | 0.064 |
| MPATS | 8.54 | 0.006 | 0.169 | 11.10 | 0.002 | 0.209 | 0.28 | 0.599 | 0.007 |
| SCS | 0.88 | 0.354 | 0.020 | 7.68 | 0.008 | 0.155 | 0.29 | 0.592 | 0.007 |

*Note.* $\eta_p^2$ = partial $\eta^2$.

smartphone use level of mindfulness group was significantly lower in post-intervention, as compared to that of pre-intervention ($p < .001$). Hypothesis 1 was verified. See Fig 2.

Repeated measure ANOVA of self-control revealed a significant time by group interaction effect, $F(1, 42) = 7.68$, $p = .008$, $\eta_p^2 = .155$, and within-subject difference of self-control across time was not significant, $F(1, 42) = 0.88$, $p = .354$, $\eta_p^2 = .020$. Furthermore, between-group difference of self-control was not significant, $F(1, 42) = .29$, $p = .592$, $\eta_p^2 = .007$. Further simple effect analysis indicated that the self-control level of mindfulness group was significantly higher in post-intervention, as compared to that of pre-intervention ($p = .012$). Hypothesis 2 was verified. See Fig 3.

## 3.4 Mediating effect analysis results

From Table 4, it can be found that the total effect of the group on post-intervention MPATS was significant ($t = .61$, 95% *CI* [1.308, 10.266]); SCS change had a significant predictive effect on post-intervention MPATS ($t = -5.44$, 95% *CI* [-0.961, -0.440]). Further analysis found that after the mediating variable entered the equation, the indirect effect Bootstrap 95% *CI* [0.490, 7.216] did not contain 0, which indicated that self-control had a mediating effect between mindfulness intervention and problematic smartphone use; besides, the direct effect Bootstrap 95% *CI* [-0.700, 6.445] contained 0, which indicated that the influence of mindfulness intervention on problematic smartphone use was completely explained by the mediating effect of self-control, that is, self-control had a complete mediating effect between mindfulness intervention and problematic smartphone use. Hypothesis 3 was verified.

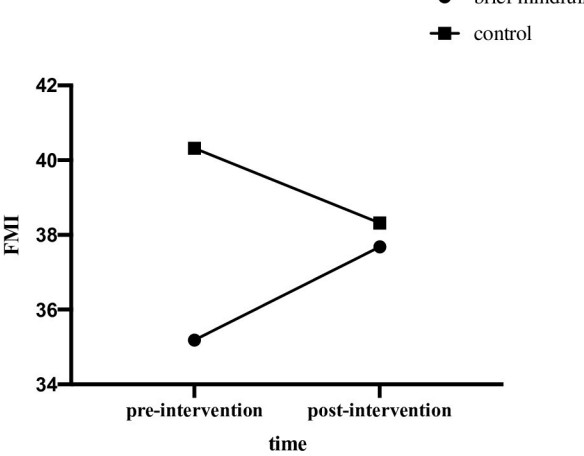

**Fig 1. FMI scores of two groups in pre- and post- intervention.**

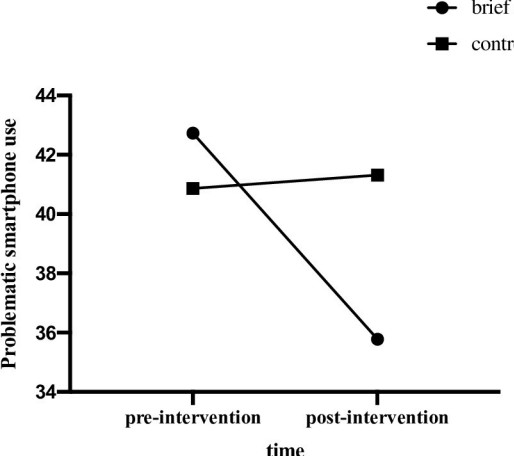

**Fig 2. Problematic smartphone use scores of two groups in pre- and post- intervention.**

## 4 Discussion

Mindfulness intervention is a method related to the training of attention ability, which can promote the individuals' self-control ability and make them live in the present moment [40]. More and more researchers choose to use brief single-session interventions instead of long-term interventions to examine the effects of mindfulness interventions [41]. However, a few studies have explored the effect of brief mindfulness interventions on problematic smartphone use prevention or its mediating mechanism. Thus, we examined the effects of a brief mindfulness intervention on problematic smartphone use and investigated whether this effect is mediated by self-control.

In general, this study found that brief mindfulness intervention can not only directly alleviate the level of problematic smartphone use among college students, but also alleviate it through improving their self-control ability. Results support all three hypotheses.

First, we predicted that a 30-min single-session mindfulness intervention would increase levels of state mindfulness, as measured by FMI, compared with the news recording. Our results yielded support for this hypothesis. On the one hand, FMI score at post-intervention

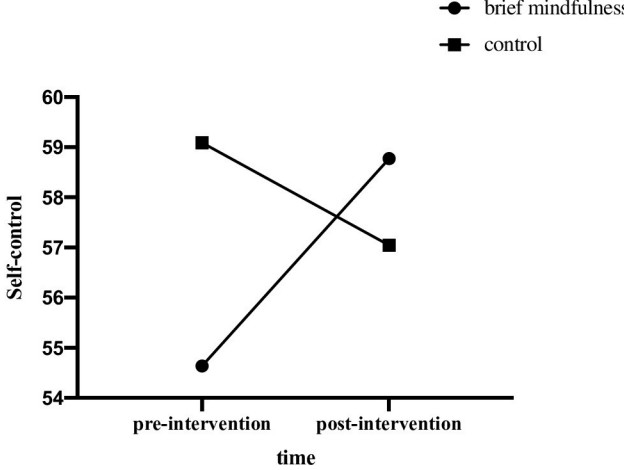

**Fig 3. Self-control scores of two groups in pre- and post- intervention.**

**Table 4. Summaries of PROCESS mediating effect test.**

| dependent variable: post-intervention MPATS | coefficient/effect | boot se | 95% CI LL | 95% CI UL |
|---|---|---|---|---|
| independent variable: | | | | |
| sex | -0.12 | 1.80 | -3.772 | 3.535 |
| age | -0.22 | 0.62 | -1.473 | 1.030 |
| pre-intervention MPATS | 0.97 | 0.07 | 0.826 | 1.118 |
| ΔSCS | -0.70 | 0.13 | -0.961 | -0.440 |
| direct effect | 2.87 | 1.76 | -0.700 | 6.445 |
| indirect effect | 2.91 | 1.64 | 0.490 | 7.216 |
| total effect | 5.78 | 2.21 | 1.308 | 10.266 |

*Note. CI* = confidence interval; Group 1 = mindfulness group, group 2 = control group.

were greater in the mindfulness intervention condition compared with pre-intervention, which is consistent with our prediction and provides additional support for previous studies that used a brief single-session mindfulness intervention to induce a state of mindfulness [39]. On the other hand, the interaction effect of time by the group was significant, and the simple effect analysis shows that, compared with the control group, the level of mindfulness of college students after a 30-minute training is significantly improved, that is, a brief mindfulness intervention can effectively improve an individual's mindfulness level.

Secondly, we found that a brief mindfulness intervention would help to alleviate the level of problematic smartphone use among college students, that is, after a brief mindfulness intervention, college students can alleviate their craving and attention to addictions, thereby blocking the relationship between addictions and problematic use behaviors. This result validated the positive impact of a brief mindfulness intervention on negative events, again. Khanna proposed that problematic use behaviors were caused by a sense of emptiness which caused individuals to satisfy themselves through substances (e.g., alcohol, nicotine, or smartphones), which could be seen as emotional disorders because of "lacking mindfulness" [42]. After being intervened, most college students can treat smartphones objectively, and gradually remain unresponsive to their desire for smartphones. The specific manifestations are: spend less time on smartphones and the alleviation importance of smartphones to life, that is, the level of problematic smartphone use is weakened.

This study also found that a brief mindfulness intervention would improve the self-control ability of college students, that is, it could enhance the activity of the self-control system and avoid the influence of psychological and behavioral impulses, which is consistent with previous studies [22]. Friese analyzed from the perspective of self-depletion and found that a one-time brief mindfulness intervention can restore self-depletion, thereby improving self-control ability [23]. The benefits of mindfulness are often conceptualized in terms of self-control [43]. In fact, individuals are required to maintain their attention on the present, and deal with distractions caused by other thoughts in the process of mindfulness training. Therefore, mindfulness training can also be regarded as self-control training for attention and concentration [44]. Individuals' self-control is repeatedly involved and practiced in this process [45], so that their self-control ability is significantly improved. These explain the reason why a brief mindfulness intervention improves college students' self-control.

Finally, we revealed the mechanism of a brief mindfulness intervention on problematic smartphone use among college students. We found that self-control played a completely mediating role in the process of brief mindfulness intervention alleviating the level of problematic smartphone use, that is, brief mindfulness interventions can not only directly alleviate the level

of problematic smartphone use among college students, but also indirectly alleviate it through self-control. The proportional analysis of the direct and indirect effects showed that the indirect effect of a mindfulness intervention on problematic smartphone use was dominant, which accounted for 50.3% of the total effect. This result reveals the relationship between mindfulness intervention and problematic smartphone use from the perspective of non-judgment and non-reaction, which is more consistent with the reality of college students' psychology and behavior and is also consistent with the "attention control" theory proposed by Diamond [46]. Brand proposed the interaction of Person-Affect-Cognition-Execution (I-PACE) model in 2016 [47], which aimed to describe the psychological and neurobiological processes underlying the development and maintenance of addictive behaviors. On this basis, Brand pointed out that problematic use behaviors were associated with diminished self-control ability [48]. In problematic use behaviors, such as problematic smartphone use, he proposed that a diminished level of self-control ability was a vulnerability factor for the problematic use behavior and acted as a mediating variable of the relationship between affective responses to certain triggering stimuli (e.g., negative moods) and decisions to engage in specific behaviors. In addition, situation-specific self-control (when being confronted with addiction-related stimuli) may decrease over time as a consequence of problematic use behaviors. Thus, self-control ability is regarded as the bridge between psychology and behavior. Mindfulness emphasizes the psychological characteristics of not judging the past and not worrying about the future [49]. If college students can be non-judgmental and non-reactive, they will pay more attention to current tasks and alleviate negative events, or they will be immersed in negative events and experiences. This explains why self-control completely mediated the influence of brief mindfulness intervention on problematic smartphone use.

Several limitations of the present study must be considered. First, although the indirect effect was significant, our sample size was relatively small, which might result in unstable estimation of the mediating effects. In addition, although the mediating effect of self-control was examined, we did not investigate whether there were potential mediating or moderating variables such as trait mindfulness, age or grade. In future research, the mediation effect should be further replicated using a larger sample size, and it should be analyzed whether the effect of brief mindfulness intervention on problematic smartphone use would be mediated or moderated by other variables.

## 5 Conclusion

In summary, the present study provided initial evidence supporting the application of a brief mindfulness intervention in problematic smartphone use and self-control for college students. Brief mindfulness interventions not only appeared to be adaptive to college students' self-control, but it also seemed to alleviate the level of problematic smartphone use. More importantly, self-control played a mediating role in the process of mindfulness intervention alleviating the level of problematic smartphone use.

## Supporting information

**S1 Data.**
(SAV)

## Acknowledgments

The authors thank Beijing Sport University for providing the psychological laboratories and participants. Also, the authors thank Mr. Yu Song and Dr. Nan Zhang for data collection at the initial stage of this study.

## Author Contributions

**Conceptualization:** Fengbo Liu, Zhongqiu Zhang.

**Data curation:** Fengbo Liu, Shuqiang Liu.

**Investigation:** Zhantao Feng.

**Writing – original draft:** Fengbo Liu.

**Writing – review & editing:** Fengbo Liu.

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
