## [Decision Letter · Decision Letter 0]

3 Aug 2021

PONE-D-21-19862

Effectiveness of Brief Mindfulness Intervention for College Students’ Smartphone Addiction: The mediating Role of Self-control

PLOS ONE

Dear Dr. Fengbo,

Thank you for submitting your manuscript to PLOS ONE. After careful consideration, we feel that it has merit but does not fully meet PLOS ONE’s publication criteria as it currently stands. Therefore, we invite you to submit a revised version of the manuscript that addresses the points raised during the review process.

In this study, correlation analyses, t-tests and ANOVAs were conducted using the subscores as well as the total scores of MPATS and SCS, but the authors made no reference to the results of those subscores in Results and Discussion sections. Moreover, the authors did not use any of those subscores in a mediation analysis. Therefore, I recommend the authors to use only the total scores of MPATS and SCS as the problematic smartphone use score and the self-control ability score, respectively. In that case, the removal of the subscore results of MPATS and SCS from Table 1-3 is also needed. Besides, it would be better to list not only pre-intervention variables but also post-intervention variables and demographic variables (i.e., gender and age) in Table 1.

We look forward to receiving your revised manuscript.

Kind regards,

Hirokazu Taniguchi, Ph.D.

Academic Editor

PLOS ONE

2. Please provide additional details regarding participant consent. In the Methods section, please ensure that you have specified (1) whether consent was informed and (2) what type you obtained (for instance, written or verbal). If your study included minors, state whether you obtained consent from parents or guardians. If the need for consent was waived by the ethics committee, please include this information.

 “This study was supported by a grant (19dz1200700) from the Shanghai Science and Technology Committee. It was also supported by Beijing Sport University.

Please include this amended Role of Funder statement in your cover letter; we will change the online submission form on your behalf."

Reviewers' comments:

Reviewer's Responses to Questions

**Comments to the Author**

1. Is the manuscript technically sound, and do the data support the conclusions?

Reviewer #1: Partly

Reviewer #2: Partly

2. Has the statistical analysis been performed appropriately and rigorously? 

Reviewer #1: Yes

Reviewer #2: Yes

3. Have the authors made all data underlying the findings in their manuscript fully available?

Reviewer #1: Yes

Reviewer #2: Yes

4. Is the manuscript presented in an intelligible fashion and written in standard English?

Reviewer #1: No

Reviewer #2: Yes

5. Review Comments to the Author

Reviewer #1: 1. The abstract should indicate whether the recruitment is random.

2. Smartphone addiction is a worldwide problem, and it is better to mention the global trend of it in the introduction section.

3. "Psychotherapy is defined as the treatment of mental health symptoms or disorders or problems of living, and/or facilitation of personal growth by psychological means", the source should be provided here.

4. "Although previous studies have preliminarily confirmed the effect of a mindfulness intervention on smartphone addiction, its mechanism has not been fully revealed", more recent research is needed here to further clarify your conclusions. What are the existing mechanisms and what needs to be supplemented?

5. What are the research gaps and significance in your study? It should be stated separately and clearly.

6. Please use more accurate intervention terms, such as "at baseline" instead of "before the experiment proper".

7. Why is there no separate table reporting demographic information in the results section? You also need to clarify whether the demographic information has potential risk of bias.

8. You'd better check whether the statements in the background and discussion sections have sources. And please update your citations, citing the latest research results as much as possible.

Reviewer #2: Thank you for the opportunity to review this manuscript. This manuscript examined associations between mindfulness, self-control, and smartphone addiction among Chinese college students. The strengths of this manuscript include a strong statistical approach, and a timely and interesting research topic. The manuscript is also generally well-written. I would suggest the authors consider the following comments to improve this manuscript's quality:

Abstract:

This sentence states, "We examined the effects of a brief mindfulness intervention on smartphone addiction and

investigated if this effect is moderated by self-control." However the authors undertook a mediation analysis. Please clarify.

Introduction:

I would argue the rationale for hypothesis 1 and Hypothesis 2 are lacking. The authors suggest and credit prior research suggesting mindfulness intervention can alleviate smartphone addiction. Then, what does Hypothesis 1 add to the literature, beyond a replication in a new sample? Similarly, the authors state mindfulness interventions can improve self-control as part of their literature review, but their hypothesis suggests mindfulness training will improve self-control in the sample. Wouldn't we assume this already, based on what is stated we know? A clearer statement of what this study would add to the literature would help prove its worth.

Methods:

It appears Chronbach's alpha for the MPATS was .65 for the pre-intervention intervention assessment, representing poor internal consistency. Why do the authors think this is so?

Please state Chronbach's alpha for the SCS.

Is a sample size of 48 enough to detect a mediated effect? Even for a reported large effect size for mindfulness, this appears a low size to detect a complicated effect such as mediation. Was the power analysis undertaken with the mediation analysis in mind, specifically? And does g*power handle mediation? Clarification that the power is appropriate for mediation is warranted, as there is possibility of Type 1 error. The authors may find the following references useful:

Schoemann, A. M., Boulton, A. J., & Short, S. D. (2017). Determining power and sample size for simple and complex mediation models. Social Psychological and Personality Science, 8(4), 379-386.

Fritz, M. S., & MacKinnon, D. P. (2007). Required sample size to detect the mediated effect. Psychological science, 18(3), 233-239.

Discussion:

5th paragraph of the discussion mentions self-control in terms of inhibiting 'animal impulses' and 'deviant behavior,' which is out of scope of the current topic. Smartphone use is quite normalized, with many people self-reporting 'addiction' to their devices, suggesting this behavior is likely not entirely abnormal or 'deviant.' The authors would improve their work by fitting in their findings within more recent literature of addictive behaviors, such as below:

Brand, M., Wegmann, E., Stark, R., Müller, A., Wölfling, K., Robbins, T. W., & Potenza, M. N. (2019). The Interaction of Person-Affect-Cognition-Execution (I-PACE) model for addictive behaviors: Update, generalization to addictive behaviors beyond internet-use disorders, and specification of the process character of addictive behaviors. Neuroscience & Biobehavioral Reviews, 104, 1-10.

Minor:

Some of the grammar and syntax throughout the paper, especially in the introduction section, could be improved. Some sentences read awkward to the reader.

The phrases of "smartphone addiction" or referencing smartphones as an addictive disorder could be rephrased to something like "problematic smartphone use,' given the debate in the field whether this behavior can be classified as a true 'addiction.'

6. PLOS authors have the option to publish the peer review history of their article (what does this mean?). If published, this will include your full peer review and any attached files.

Reviewer #1: No

Reviewer #2: **Yes: **Timothy Regan

---

## [Author Response · Author response to Decision Letter 0]

18 Oct 2021

Responds to the reviewer’s comments:

In this study, correlation analyses, t-tests and ANOVAs were conducted using the subscores as well as the total scores of MPATS and SCS, but the authors made no reference to the results of those subscores in Results and Discussion sections. Moreover, the authors did not use any of those subscores in a mediation analysis. Therefore, I recommend the authors to use only the total scores of MPATS and SCS as the problematic smartphone use score and the self-control ability score, respectively. In that case, the removal of the subscore results of MPATS and SCS from Table 1-3 is also needed. Besides, it would be better to list not only pre-intervention variables but also post-intervention variables and demographic variables (i.e., gender and age) in Table 1.

Response: It is really true as Reviewer said that we made no results of subscores in Results and Discussion sections because they were not the focus of our research. Thus, as you suggested, we have removed all subscore results in our paper.

Moreover, demographic variables were compared in table 1. Thus, as you suggested, we have listed pre-intervention variables, post-intervention variables and demographic variables in Table 2. 

Reviewer 1:

1. The abstract should indicate whether the recruitment is random.

Response: It is really true as Reviewer suggested that we should indicate whether the recruitment is random, and now we have made correction in abstract according to your comments.

2. Smartphone addiction is a worldwide problem, and it is better to mention the global trend of it in the introduction section.

Response: Thank you for your comment. Now we have mentioned the global trend of smartphone addiction in the introduction section.

3. "Psychotherapy is defined as the treatment of mental health symptoms or disorders or problems of living, and/or facilitation of personal growth by psychological means", the source should be provided here.

Response: We are very sorry for our negligence, and as you suggested, the source was provided here.

4. "Although previous studies have preliminarily confirmed the effect of a mindfulness intervention on smartphone addiction, its mechanism has not been fully revealed", more recent research is needed here to further clarify your conclusions. What are the existing mechanisms and what needs to be supplemented?

Response: We are very sorry for our negligence, and we have re-written this sentence. “Most existing studies focused on the mediating effect of trait mindfulness, rather than the mediating mechanism of mindfulness intervention. Therefore, it is necessary to further supplement the mediating mechanism of mindfulness training on mobile phone addiction”.

5. What are the research gaps and significance in your study? It should be stated separately and clearly.

Response: It is really true as Reviewer suggested that the research gaps and significance should be stated separately and clearly. Thus, now we have made a supplement of these sections and re-written our hypothesis in the end of the Introduction.

6. Please use more accurate intervention terms, such as "at baseline" instead of "before the experiment proper".

Response: We are very sorry for our incorrect writing, now we have replaced “before the experiment proper” with “at baseline”, and examined other mistakes.

7. Why is there no separate table reporting demographic information in the results section? You also need to clarify whether the demographic information has potential risk of bias.

Response: We are very sorry for our negligence, and the separate table reporting demographic information was showed at results section now. The demographic variables were also compared in table 1. 

8. You'd better check whether the statements in the background and discussion sections have sources. And please update your citations, citing the latest research results as much as possible.

Response: Thank you for your comment. Now, all sources were provided in the background and discussion sections. We also updated our citations, and citing the latest studies as much as possible.

Reviewer 2:

Abstract:

This sentence states, "We examined the effects of a brief mindfulness intervention on smartphone addiction and investigated if this effect is moderated by self-control." However the authors undertook a mediation analysis. Please clarify.

Response: We are very sorry for our incorrect writing, now we have replaced “moderated” with “mediated” in abstract section.

Introduction:

I would argue the rationale for hypothesis 1 and Hypothesis 2 are lacking. The authors suggest and credit prior research suggesting mindfulness intervention can alleviate smartphone addiction. Then, what does Hypothesis 1 add to the literature, beyond a replication in a new sample? Similarly, the authors state mindfulness interventions can improve self-control as part of their literature review, but their hypothesis suggests mindfulness training will improve self-control in the sample. Wouldn't we assume this already, based on what is stated we know? A clearer statement of what this study would add to the literature would help prove its worth.

Response: It is really true as Reviewer suggested that a clearer statement of what this study would add to the literature is needed. The majority of previous studies had focused on the effects of long-term mindfulness training on self-control and smartphone addiction, while a few intervention studies had explored the relationship between them for college students; thus, the adoption of brief mindfulness training in this study is purposeful and fills a missing research gap. Moreover, previous studies have examined the changes in smartphone addiction and self-control before and after mindfulness intervention. However, the mediating mechanism by which mindfulness training improves smartphone addiction has rarely been explored. Thus, now we have made a supplement of these sections and re-written our hypothesis in the end of the Introduction.

Methods:

It appears Chronbach's alpha for the MPATS was .65 for the pre-intervention intervention assessment, representing poor internal consistency. Why do the authors think this is so?

Response: Thank you for your comment. In our assessments, .65 was not the Chronbach's alpha for the MPATS, while it was the alpha score for a subscale of MPATS. As the Reviewer suggested, we have removed all subscore results in our paper. And the alpha for the MPATS was .91, which means that there was acceptable internal consistency reliability for MPATS.

Please state Chronbach's alpha for the SCS.

Response: We are very sorry for our negligence, and the Chronbach's alpha for the SCS has been supplemented now.

Is a sample size of 48 enough to detect a mediated effect? Even for a reported large effect size for mindfulness, this appears a low size to detect a complicated effect such as mediation. Was the power analysis undertaken with the mediation analysis in mind, specifically? And does g*power handle mediation? Clarification that the power is appropriate for mediation is warranted, as there is possibility of Type 1 error. The authors may find the following references useful:

Schoemann, A. M., Boulton, A. J., & Short, S. D. (2017). Determining power and sample size for simple and complex mediation models. Social Psychological and Personality Science, 8(4), 379-386.

Fritz, M. S., & MacKinnon, D. P. (2007). Required sample size to detect the mediated effect. Psychological science, 18(3), 233-239.

Response: Thank you for your comment. As you said, G*power can’t handle mediation sample size. While in our study, we used G*power to only determine the sample size of brief mindfulness training on smartphone addiction.

Moreover, in the previous questionnaire survey research, the number of subjects in the mediation effect test needs to reach a large sample size. However, in experimental research, it is impossible to complete the intervention of a large sample size. Therefore, according to previous studies (Josefsson, T. et al. (2019). Effects of Mindfulness-Acceptance-Commitment (MAC) on Sport-Specific Dispositional Mindfulness, Emotion Regulation, and Self-Rated Athletic Performance in a Multiple-Sport Population: an RCT Study. Mindfulness), we recruited 48 subjects in our research. The PROCESS was used to test the mediating effect of self-control between mindfulness intervention and smartphone addiction, The number of Bootstrap samples was 5000.

Discussion:

5th paragraph of the discussion mentions self-control in terms of inhibiting 'animal impulses' and 'deviant behavior,' which is out of scope of the current topic. Smartphone use is quite normalized, with many people self-reporting 'addiction' to their devices, suggesting this behavior is likely not entirely abnormal or 'deviant.' The authors would improve their work by fitting in their findings within more recent literature of addictive behaviors, such as below:

Brand, M., Wegmann, E., Stark, R., Müller, A., Wölfling, K., Robbins, T. W., & Potenza, M. N. (2019). The Interaction of Person-Affect-Cognition-Execution (I-PACE) model for addictive behaviors: Update, generalization to addictive behaviors beyond internet-use disorders, and specification of the process character of addictive behaviors. Neuroscience & Biobehavioral Reviews, 104, 1-10.

Response: We are very sorry for our incorrect writing, and thank you for your comment. We have re-written this part, which aims to explain more clearly about the mediating role of self-control.

Minor:

Some of the grammar and syntax throughout the paper, especially in the introduction section, could be improved. Some sentences read awkward to the reader.

Response: Thank you for your comment. We have examined the grammar and syntax throughout the paper, and re-written the most parts of introduction section in our paper.

The phrases of "smartphone addiction" or referencing smartphones as an addictive disorder could be rephrased to something like "problematic smartphone use,' given the debate in the field whether this behavior can be classified as a true 'addiction.'

Response: Thank you for your comment. As you suggested, it is debatable whether this behavior can be classified as a true 'addiction'. And the tool we use can only measure the tendency of smartphone addiction. Therefore, we have replaced “smartphone addiction” with “problematic smartphone use”.

Special thanks to you for all your good comments!

We tried our best to improve the manuscript and made some changes in the manuscript. These changes will not influence the content and framework of the paper. And here we did not list the changes but marked in red in revised paper.

We appreciate for Editors/Reviewers’ warm work earnestly, and hope that the correction will meet with approval.

Once again, thank you very much for your comments and suggestions.

---

## [Decision Letter · Decision Letter 1]

31 Jan 2022

PONE-D-21-19862R1Effectiveness of Brief Mindfulness Intervention for College Students’ Problematic Smartphone Use: The mediating Role of Self-controlPLOS ONE

Dear Dr. Fengbo,

Thank you for submitting your manuscript to PLOS ONE. After careful consideration, we feel that it has merit but does not fully meet PLOS ONE’s publication criteria as it currently stands. Therefore, we invite you to submit a revised version of the manuscript that addresses the points raised during the review process.

As both reviewers pointed out, if the authors did not carry out a power analysis to calculate the sample size necessary to achieve a specific level of power (1-β = 0.8) in their mediation model, they had better mention it for accuracy purposes. I understand the percentile bootstrap CI is more powerful than Sobel test, indicates less Type I error inflation in smaller samples (*n* < 100) when there is no indirect effect (*ab* is equal to zero), and provides better CI coverage than the bias-corrected bootstrap CI (cf. Hayes & Scharkow, 2013). As an aside, generally speaking, when sample size is small, statistical power (1-β) becomes low, and Type II error becomes high. Although the sample size was relatively small in this study, the indirect effect was significant in the percentile bootstrap CI. If necessary, the authors might be able to try a "post hoc" power analysis, for example, using Schoemann's (2017) application for Monte Carlo power analysis for mediation models. As the reviewer #3 suggested, Tables 3 and 4 are unnecessary because the results of the two-way repeated measures ANOVAs have been explained in the text and displayed in Figures 1-3. I think it would be better to use a figure rather than a table, that is Table 5, to present the results of the mediating effect analysis. Before converting Table 5 to a figure, I recommend the authors to check again whether the values of the direct, indirect, and total effects are correct in Table 5.

We look forward to receiving your revised manuscript.

Kind regards,

Hirokazu Taniguchi, Ph.D.

Academic Editor

PLOS ONE

Reviewers' comments:

Reviewer's Responses to Questions

**Comments to the Author**

1. If the authors have adequately addressed your comments raised in a previous round of review and you feel that this manuscript is now acceptable for publication, you may indicate that here to bypass the “Comments to the Author” section, enter your conflict of interest statement in the “Confidential to Editor” section, and submit your "Accept" recommendation.

Reviewer #2: (No Response)

Reviewer #3: (No Response)

2. Is the manuscript technically sound, and do the data support the conclusions?

Reviewer #2: Yes

Reviewer #3: No

3. Has the statistical analysis been performed appropriately and rigorously? 

Reviewer #2: Yes

Reviewer #3: No

4. Have the authors made all data underlying the findings in their manuscript fully available?

Reviewer #2: Yes

Reviewer #3: No

5. Is the manuscript presented in an intelligible fashion and written in standard English?

Reviewer #2: Yes

Reviewer #3: No

6. Review Comments to the Author

Reviewer #2: I would recommend stating the inclusion criteria.

Lan reported a medium-to-large effect size of the mindfulness effect for college students [19] - I recommend more clearly stating this effect. What is meant by mindfulness, i.e. trait, intervention, something else?

I would recommend stating the power analysis was not undertaken with the mediation effect in mind. The authors need to be careful in their wording, as their small sample size remains a limitation of the study.

Reviewer #3: have fully addressed the comment: not yet

technically sound: no, see my comment below

statistical analysis: no, see my comment below

data fully available: i can't find it, so 'no'

English: no. need proof-reading

specific comments:

line 65-66: the statement is odd. mindfulness intervention could increase one's trait mindfulness. so, if trait mindfulness is found negatively associated with problematic smartphone use. we could implement mindfulness-based intervention to increase one's trait mindfulness.

line 66-67"it is necessary to further supplement the mediating mechanism of mindfulness intervention on mobile phone addiction": 1)mediate or moderate? in the response to previous reviewer, the authors said it should 'moderate'. 2) seems that it is an interventional studies, the authors should look into the effect of mindfulness intervention on problematic smartphone. I don't understand why such study design is for a research question about mediating effect. previous reviewers also asked similar question.

line 87-88" how to define long term training? no elaboration on why need to look into brief mindfulness training. I am not sure whether 'brief' =short duration, single modality or else.

line 91-94'Clearly, there could be potential confounding factors arising from prolonged mindfulness interventions, such as improved interpersonal relationship [30], which makes it difficult for singling out mindfulness practice as the cause of improved problematic smartphone use.': this argument is certainly not sound. 1) we have methodological ways to minimize potential confounder. 2) no definition of brief mindfulness intervention. it is certainly not a good justification to introduce 'brief' mindfulness training.

intervention: no evidence to support the design of the intervention - 30min audio-taped mindfulness.

the sample size calculation is for comparing difference but not for mediating analysis. 48 samples are certainly insufficient to do a mediating analysis with a rule of one IDV needs 15 samples. no justification is given in the section 'sample size' for the sample for mediating analysis.

With table 4, table 3 is not needed.

7. PLOS authors have the option to publish the peer review history of their article (what does this mean?). If published, this will include your full peer review and any attached files.

Reviewer #2: **Yes: **Timothy Regan

Reviewer #3: No

---

## [Author Response · Author response to Decision Letter 1]

23 Oct 2022

Responds to the reviewer’s comments:

As both reviewers pointed out, if the authors did not carry out a power analysis to calculate the sample size necessary to achieve a specific level of power (1-β = 0.8) in their mediation model, they had better mention it for accuracy purposes. I understand the percentile bootstrap CI is more powerful than Sobel test, indicates less Type I error inflation in smaller samples (n < 100) when there is no indirect effect (ab is equal to zero), and provides better CI coverage than the bias-corrected bootstrap CI (cf. Hayes & Scharkow, 2013). As an aside, generally speaking, when sample size is small, statistical power (1-β) becomes low, and Type II error becomes high. Although the sample size was relatively small in this study, the indirect effect was significant in the percentile bootstrap CI. If necessary, the authors might be able to try a "post hoc" power analysis, for example, using Schoemann's (2017) application for Monte Carlo power analysis for mediation models. As the reviewer #3 suggested, Tables 3 and 4 are unnecessary because the results of the two-way repeated measures ANOVAs have been explained in the text and displayed in Figures 1-3. I think it would be better to use a figure rather than a table, that is Table 5, to present the results of the mediating effect analysis. Before converting Table 5 to a figure, I recommend the authors to check again whether the values of the direct, indirect, and total effects are correct in Table 5.

Response: Thank you for your comments. As previous studies showed, G*power can’t handle mediation sample size. While in experimental research, it is impossible to complete the intervention of a large sample size. Thus, we have mentioned it in limitation part for accuracy purposes. 

Secondly, as reviewer #3 suggested, Tables 3 and 4 were deleted. 

Finally, it may not be better to convert Table 5 into a figure, because the table showed 95% CI and other data, which were the key data for the mediating effect test. If necessary, a figure can be added on the basis of Table 5, but there may be suspicion of the duplication.

Reviewer 2:

I would recommend stating the inclusion criteria.

Lan reported a medium-to-large effect size of the mindfulness effect for college students [19] - I recommend more clearly stating this effect. What is meant by mindfulness, i.e. trait, intervention, something else?

Response: Thank you for your comment. Lan reported a medium-to-large effect size of group mindfulness-based intervention for college students’ problematic smartphone use. Now we have made it more clearly.

I would recommend stating the power analysis was not undertaken with the mediation effect in mind. The authors need to be careful in their wording, as their small sample size remains a limitation of the study.

Response: It is really true as Reviewer suggested that we need to be careful in our wording. Thus, now we have rewritten the limitation part in the end of the Discussion.

Reviewer 3:

specific comments:

line 65-66: the statement is odd. mindfulness intervention could increase one's trait mindfulness. so, if trait mindfulness is found negatively associated with problematic smartphone use. we could implement mindfulness-based intervention to increase one's trait mindfulness.

Response: Thank you for your comment. As reviewer said, we could implement mindfulness-based intervention to increase one's trait mindfulness, and then, may reduce the problematic smartphone use. While at this point, the level of trait mindfulness is regarded as a mediating variable. But our study focused on the mediating effect of self-control. Thus, it can be further discussed in future study that whether the effect of brief mindfulness intervention on problematic smartphone use would be mediated by trait mindfulness, and now we have rewritten the limitation part in the end of the Discussion.

line 66-67"it is necessary to further supplement the mediating mechanism of mindfulness intervention on mobile phone addiction": 1)mediate or moderate? in the response to previous reviewer, the authors said it should 'moderate'. 2) seems that it is an interventional studies, the authors should look into the effect of mindfulness intervention on problematic smartphone. I don't understand why such study design is for a research question about mediating effect. previous reviewers also asked similar question.

Response: We are very sorry for our incorrect writing. One of our purposes was to analyze the effect of mindfulness intervention on problematic phone use, while previous studies had shown that mindfulness intervention can significantly alleviate the level of problematic smartphone use. But, how this effect was produced need to be further explored, thus we further supplemented its mediating mechanism.

line 87-88" how to define long term training? no elaboration on why need to look into brief mindfulness training. I am not sure whether 'brief' =short duration, single modality or else.

Response: Thank you for your comment. There are many long-term intervention methods based on mindfulness, such as MBCT or MBSR, which are all 8-week long term training. In our study, brief means one-time (30 ~ 40 minutes each time). Previous studies provided evidence for this (Schumer, M. C., et al. Brief Mindfulness Training for Negative Affectivity: A Systematic Review and Meta-Analysis). 

line 91-94'Clearly, there could be potential confounding factors arising from prolonged mindfulness interventions, such as improved interpersonal relationship [30], which makes it difficult for singling out mindfulness practice as the cause of improved problematic smartphone use.': this argument is certainly not sound. 1) we have methodological ways to minimize potential confounder. 2) no definition of brief mindfulness intervention. it is certainly not a good justification to introduce 'brief' mindfulness training.

Response: Thank you for your comments. As reviewer said, we have methodological ways to minimize the potential confounder, while there are still other potential confounders of prolonged interventions which can not be minimized, such as habit formation, et al., while these can be avoided by using brief training. We have put forward this point in the article.

intervention: no evidence to support the design of the intervention - 30min audio-taped mindfulness.

Response: We are very sorry for our negligence. In our study, brief intervention of 30 min audio-taped was used, and relevant references confirmed that one-term audio-taped intervention was effective, which have been added to the article.

the sample size calculation is for comparing difference but not for mediating analysis. 48 samples are certainly insufficient to do a mediating analysis with a rule of one IDV needs 15 samples. no justification is given in the section 'sample size' for the sample for mediating analysis.

Response: Thank you for your comment. As previous studies showed, G*power is for comparing difference but not for mediating analysis. However, the bootstrap method used in this study was effective, and previous studies have provided evidence for this. Moreover, in experimental research, it is impossible to complete the intervention of a large sample size. Thus, we have mentioned it in limitation part for accuracy purposes.

With table 4, table 3 is not needed.

Response: Thank you for your comment. As you suggested, we have deleted table 3 and table 4.

Special thanks to you for all your good comments!

We tried our best to improve the manuscript and made some changes in the manuscript. These changes will not influence the content and framework of the paper. And here we did not list the changes but marked in red in revised paper.

We appreciate for Editors/Reviewers’ warm work earnestly, and hope that the correction will meet with approval.

Once again, thank you very much for your comments and suggestions.

---

## [Decision Letter · Decision Letter 2]

22 Nov 2022

PONE-D-21-19862R2Effectiveness of Brief Mindfulness Intervention for College Students’ Problematic Smartphone Use: The mediating Role of Self-controlPLOS ONE

Dear Dr. Liu,

Thank you for submitting your manuscript to PLOS ONE. After careful consideration, we feel that it has merit but does not fully meet PLOS ONE’s publication criteria as it currently stands. Therefore, we invite you to submit a revised version of the manuscript that addresses the points raised during the review process.

The authors should italicize statistical symbols and add the value of degrees of freedom. In the results of independent-sample t-tests, the *p*-value of FMI seems wrong. Table 3 does not include "group," one of the independent variables. The authors should insert a space between words or after commas in many sentences, such as L28, L180, L197, L311, L315, L317, and L318. It would be better to change "fourteen items" to "14 items" in L142 and "applied by" to "applied to" in L110.

We look forward to receiving your revised manuscript.

Kind regards,

Hirokazu Taniguchi, Ph.D.

Section Editor

PLOS ONE

Journal Requirements:

Reviewers' comments:

Reviewer's Responses to Questions

**Comments to the Author**

1. If the authors have adequately addressed your comments raised in a previous round of review and you feel that this manuscript is now acceptable for publication, you may indicate that here to bypass the “Comments to the Author” section, enter your conflict of interest statement in the “Confidential to Editor” section, and submit your "Accept" recommendation.

Reviewer #2: All comments have been addressed

Reviewer #3: (No Response)

2. Is the manuscript technically sound, and do the data support the conclusions?

Reviewer #2: Partly

Reviewer #3: Partly

3. Has the statistical analysis been performed appropriately and rigorously? 

Reviewer #2: Yes

Reviewer #3: Yes

4. Have the authors made all data underlying the findings in their manuscript fully available?

Reviewer #2: Yes

Reviewer #3: No

5. Is the manuscript presented in an intelligible fashion and written in standard English?

Reviewer #2: Yes

Reviewer #3: No

6. Review Comments to the Author

Reviewer #2: I would recommend stating the mediation effect as more of a preliminary finding in need of further replication using a larger sample. When the sample size is this small, applying mediation is more of an 'exploratory' route.

I would also recommend stating what the participants actually did during the mindfulness exercise? Activities, meditation, etc. are mentioned but not described.

Reviewer #3: for the ANOVA result, you shall present it with table.

it is not a 2x2 factorial design. I guess it is a quasi-experimental only.

given the above study design, you should use general linear model instead of 2-way repeated measure ANOVA for data analysis. You better consult a statistician for the correct test used.

7. PLOS authors have the option to publish the peer review history of their article (what does this mean?). If published, this will include your full peer review and any attached files.

Reviewer #2: **Yes: **Timothy Regan

Reviewer #3: No

---

## [Author Response · Author response to Decision Letter 2]

28 Nov 2022

Responds to the reviewer’s comments:

The authors should italicize statistical symbols and add the value of degrees of freedom. In the results of independent-sample t-tests, the p-value of FMI seems wrong. Table 3 does not include "group," one of the independent variables. The authors should insert a space between words or after commas in many sentences, such as L28, L180, L197, L311, L315, L317, and L318. It would be better to change "fourteen items" to "14 items" in L142 and "applied by" to "applied to" in L110.

Response: Thank you for your comments. Firstly, we are very sorry for our incorrect writing, and we have italicized statistical symbols, added the value of degrees of freedom, and rewritten the p-value of FMI. Secondly, we have inserted a space between words or after commas in above sentences, and we have also changed "fourteen items" to "14 items" and "applied by" to "applied to". Finally, in table 3, the result of “group” was same as “direct effect”, thus it did not include "group" to avoid duplication. In addition, according to the suggestion of Reviewer 3, we presented the ANOVA results in table 3. 

Reviewer 2:

I would recommend stating the mediation effect as more of a preliminary finding in need of further replication using a larger sample. When the sample size is this small, applying mediation is more of an 'exploratory' route.

Response: It is really true as Reviewer suggested that our sample size was relatively small. Thus, in future research, the mediation effect should be further replicated using a larger sample size. Now we have rewritten the limitation part in the end of the Discussion.

I would also recommend stating what the participants actually did during the mindfulness exercise? Activities, meditation, etc. are mentioned but not described.

Response: Thank you for your comments. As reviewer recommend, we have stated what the participants actually did during the mindfulness exercise in “2.3 Procedure” part.

Reviewer 3:

for the ANOVA result, you shall present it with table.

Response: Thank you for your comments. As reviewer recommend, we have presented the ANOVA results in table 3.

it is not a 2x2 factorial design. I guess it is a quasi-experimental only. given the above study design, you should use general linear model instead of 2-way repeated measure ANOVA for data analysis. You better consult a statistician for the correct test used.

Response: Thank you for your comment. In our study, the group (mindfulness group vs. control group) was a between-subject factor, and the time (pre-intervention vs. post-intervention) was a within-subject factor, thus it could been seen as a 2×2 mixed factorial design. And 2-way repeated measure ANOVA should be used for data analysis, and previous studies provided evidence for this (Zhang, C.-Q., Si, G., Duan, Y., Lyu, Y., Keatley, D. A., & Chan, D. K. C. (2016). The effects of mindfulness training on beginners’ skill acquisition in dart throwing: A randomized controlled trial. Psychology of Sport and Exercise, 22, 279–285.).

We tried our best to improve the manuscript and made some changes in the manuscript. These changes will not influence the content and framework of the paper. And here we did not list the changes but marked in red in revised paper.

We appreciate for Editors/Reviewers’ warm work earnestly, and hope that the correction will meet with approval.

Once again, thank you very much for your comments and suggestions.

---

## [Decision Letter · Decision Letter 3]

12 Dec 2022

Effectiveness of Brief Mindfulness Intervention for College Students’ Problematic Smartphone Use: The mediating Role of Self-control

PONE-D-21-19862R3

Dear Dr. Liu,

We’re pleased to inform you that your manuscript has been judged scientifically suitable for publication and will be formally accepted for publication once it meets all outstanding technical requirements.

Kind regards,

Hirokazu Taniguchi, Ph.D.

Section Editor

PLOS ONE

Additional Editor Comments (optional):

In reporting the mediating effect analysis, the authors should indicate not only the effect of SCS change on post-intervention MPATS (one of the components of the mediating effect) but also the effect of group on SCS change (the other components).

As reviewer #3 has commented, it is worthwhile to use the general linear model and compare the model results with the results of the ANOVA.

Reviewers' comments:

Reviewer's Responses to Questions

**Comments to the Author**

1. If the authors have adequately addressed your comments raised in a previous round of review and you feel that this manuscript is now acceptable for publication, you may indicate that here to bypass the “Comments to the Author” section, enter your conflict of interest statement in the “Confidential to Editor” section, and submit your "Accept" recommendation.

Reviewer #3: All comments have been addressed

2. Is the manuscript technically sound, and do the data support the conclusions?

Reviewer #3: Yes

3. Has the statistical analysis been performed appropriately and rigorously? 

Reviewer #3: Yes

4. Have the authors made all data underlying the findings in their manuscript fully available?

Reviewer #3: Yes

5. Is the manuscript presented in an intelligible fashion and written in standard English?

Reviewer #3: Yes

6. Review Comments to the Author

Reviewer #3: just one point about the study design. properly, it is a quasi-experimental design with manipulation and control but no randomization.

7. PLOS authors have the option to publish the peer review history of their article (what does this mean?). If published, this will include your full peer review and any attached files.

Reviewer #3: No

---

## [Editor Report · Acceptance letter]

14 Dec 2022

PONE-D-21-19862R3 

Effectiveness of Brief Mindfulness Intervention for College Students’ Problematic Smartphone Use: The mediating Role of Self-control 

Dear Dr. Liu:

I'm pleased to inform you that your manuscript has been deemed suitable for publication in PLOS ONE. Congratulations! Your manuscript is now with our production department. 

Kind regards, 

on behalf of

Dr. Hirokazu Taniguchi 

Section Editor

PLOS ONE